# Modality-Agnostic Self-Supervised Learning with Meta-Learned Masked Auto-Encoder

**Huiwon Jang**[A*] **Jihoon Tack**[A*] **Daewon Choi**[B] **Jongheon Jeong**[A] **Jinwoo Shin**[A]
[A]Korea Advanced Institute of Science and Technology (KAIST)  [B]Korea University
{huiwoen0516, jihoontack}@kaist.ac.kr

## Abstract

Despite its practical importance across a wide range of modalities, recent advances in self-supervised learning (SSL) have been primarily focused on a few well-curated domains, e.g., vision and language, often relying on their domain-specific knowledge. For example, *Masked Auto-Encoder* (MAE) has become one of the popular architectures in these domains, but less has explored its potential in other modalities. In this paper, we develop MAE as a unified, modality-agnostic SSL framework. In turn, we argue *meta-learning* as a key to interpreting MAE as a modality-agnostic learner, and propose enhancements to MAE from the motivation to jointly improve its SSL across diverse modalities, coined *MetaMAE* as a result. Our key idea is to view the mask reconstruction of MAE as a meta-learning task: masked tokens are predicted by adapting the Transformer meta-learner through the amortization of unmasked tokens. Based on this novel interpretation, we propose to integrate two advanced meta-learning techniques. First, we adapt the amortized latent of the Transformer encoder using gradient-based meta-learning to enhance the reconstruction. Then, we maximize the alignment between amortized and adapted latents through task contrastive learning which guides the Transformer encoder to better encode the task-specific knowledge. Our experiment demonstrates the superiority of MetaMAE in the modality-agnostic SSL benchmark (called DABS), significantly outperforming prior baselines. Code is available at https://github.com/alinlab/MetaMAE.

## 1 Introduction

Self-supervised learning (SSL), i.e., learning without human supervision, recently has demonstrated substantial success across fields including, computer vision [32, 11, 29, 47, 33, 5, 102], natural language processing (NLP) [18, 49, 55, 70], and speech recognition [2, 38, 39]. The efficacy of SSL is derived by extracting transferable knowledge from unlabeled datasets, a feature that manifests significant utility for various downstream tasks such as classification and segmentation. As a result, SSL has become an indispensable technique in real-world applications (for instance, industrial contexts like medical imaging [26]), not only improving the performance on new datasets but also reducing a significant amount of computations and costs, e.g., expert annotation poses significant costs [68, 16]. However, despite the importance of SSL in such fields, recent advancements have been predominantly focused on specific domains (e.g., images and NLP) where the majority of existing SSL frameworks on such domains require modality-specific knowledge, thereby constraining the applicability and scalability of previous works across new modalities.

To tackle this issue, we draw attention to the recent success of the Masked Auto-Encoder (MAE) framework [33], which eliminates the need for modality-specific inductive biases. Initially presented as a generative model [91, 67], the MAE models the network to reconstruct the original input

---

*Equal contributions

37th Conference on Neural Information Processing Systems (NeurIPS 2023).

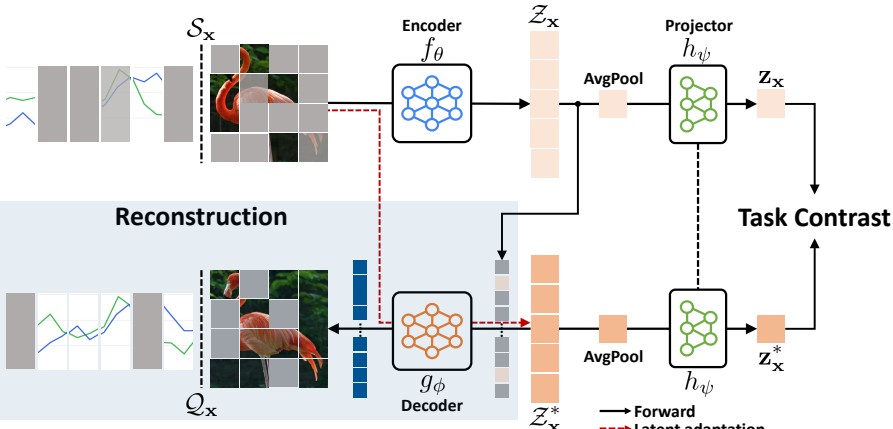

Figure 1: An overview of the proposed Meta-learned Masked Auto-Encoder (MetaMAE): we adapt the amortized latent $\mathcal{Z}_{\mathbf{x}}$ of the Transformer encoder $f_\theta$ using gradient-based meta-learning on the Transformer decoder $g_\phi$ to enhance the reconstruction, then maximize the alignment with optimized latent $\mathcal{Z}_{\mathbf{x}}^*$ to guide the Transformer encoder towards improved predictions via task contrastive learning.

signal based on the randomly masked part of the signal. Recently, the integration of MAE with the Transformer architecture [33] has resulted in a powerful SSL framework for various domains, such as vision [33, 5, 10], NLP [18, 70], and tabular [59] datasets. For instance, it has been demonstrated that BERT [18], utilizing a Transformer encoder with a linear decoder, can effectively transfer to diverse tasks for the NLP domain. On the other hand, it has been suggested that utilizing a decoder of a deep Transformer architecture is essential for applying MAE within the vision domain [106], achieving remarkable performance [33]. Building on these insights, we found that MAE is quite a promising direction for modality-agnostic SSL: our experiment demonstrates that using a deep Transformer decoder for MAE significantly outperforms previous modality-agnostic SSL frameworks (see Table 5). In this paper, we suggest to further exploit the benefits of MAE to build a unified SSL framework.

**Contribution.** We propose Meta-learned Masked Auto-Encoder (MetaMAE), a novel modality-agnostic SSL framework that leverages the power of meta-learning; see the overview in Figure 1. Our key idea is to interpret MAE as a meta-learning framework, thereby improving the generalization through the use of advanced meta-learning schemes. To be specific, we interpret the data reconstruction itself as a task, where the Transformer meta-learner is adapted through amortization of the support set (i.e., unmasked tokens) to predict the query set (i.e., masked tokens). Based on this interpretation, we propose a novel integration of two advanced meta-learning techniques to enhance MAE; namely the use of gradient-based meta-learning [22] and task contrastive learning [28, 60].

- *Latent Adaptation via Gradient-based Meta-learning*: We suggest adapting the amortized latent to better reconstruct the given support set (and the nearby tokens) through gradient-based meta-learning on the decoder [74]. Then, the optimized latent is used to condition the decoder for the query prediction. This approach generally eases the task compared to the direct reconstruction, thereby streamlining and improving the task adaptation process.

- *Task Contrastive Learning*: To further leverage this optimized latent, we suggest utilizing task contrastive learning [28, 60]. Specifically, since both the optimized and predicted latents originate from the same task, we aim to maximize their similarity while minimizing their similarity with other tasks. This prompts the Transformer encoder to produce predictions closely aligned with the optimized latent, effectively guiding the Transformer to better encode the task knowledge.

We verify the efficacy of MetaMAE through extensive evaluations on multiple data modalities from modality-agnostic SSL benchmarks (i.e., DABS 1.0 [83] and 2.0 [85]), including time-series, tabular, discrete token, multi-spectral images, speech, and multi-modal datasets. Overall, our experimental results demonstrate strong results, consistently and significantly outperforming previous modality-agnostic SSL methods in linear evaluation. For instance, measured with classification accuracy (%), MetaMAE improves the prior state-of-the-art results by $85.3 \rightarrow 89.3$ on PAMAP2 [71], $53.6 \rightarrow 69.4$ on Genomics [72], and $60.2 \rightarrow 79.8$ on LibriSpeech [66] datasets. Moreover, we also demonstrate that MetaMAE significantly improves the linear evaluation performance on cross-domain datasets, indicating the improved transfer ability of MAE through meta-learning.

## 2 Meta-Learning Modality-Agnostic Masked Auto-Encoder

In this section, we present *Meta-learned Masked Auto-Encoder (MetaMAE)*, a novel and effective modality-agnostic self-supervised learning (SSL) framework. Our key contribution is to tackle the modality-agnostic SSL problem with in-depth utilization of MAE, which was quite under-explored in the field. Based on our novel interpretation that views MAE as a meta-learning framework (in Section 2.1), we improve the transfer ability of MAE by suggesting enhanced modality-agnostic meta-learning techniques including latent optimization-based meta-learning and task contrastive learning (in Section 2.2). Our framework is visually depicted in Figure 1, and the pseudo-code is provided in Algorithm 1.

**Problem setup.** We first describe the problem setup of our interest, modality-agnostic SSL. This problem aims to learn a transferable representation from the unlabeled dataset without utilizing the modality-specific inductive biases. For a given unlabeled pretrain dataset $\mathcal{D}_{\texttt{pretrain}} = \{\mathbf{x}_i\}_{i=1}^N$, where $\mathbf{x} \in \mathbb{R}^d$ represents an input sampled from a certain data-generating distribution in an i.i.d manner, our objective is to train an encoder $f_\theta$ that can linearly separate the given labeled transfer dataset drawn from a similar or the same data-generating distribution.

### 2.1 Rethinking Masked Auto-Encoder as a Meta-Learning Framework

Meta-learning [86] aims to extract and utilize the knowledge from the distribution of tasks to better solve a relevant task. This problem is typically approached by training a meta-learner that can transfer its knowledge to a task-specific model through adaptation, where the performance of the meta-learner is evaluated on the basis of how well each adapted model performs on the corresponding task. To learn such a meta-learner, a standard way is to use a set of *support set* samples to adapt the task-specific model from the meta-learner and use another disjoint set of samples, called *query set* samples to evaluate the adaptation performance [92, 82].

**Mask prediction as a modality-agnostic task.** MAE is an SSL technique that trains an autoencoder to reconstruct the original input signal with a randomly masked part of the signal. To implement such a technique, recent works utilize the Transformer architecture for the autoencoder design which is necessary for successful training. To use Transformer for MAE, the input data is broken down into non-overlapping units coined tokens (e.g., patches for images, and words for languages) where such tokens are divided into two disjoint sets (unmasked and masked) for MAE modeling, i.e., the Transformer autoencoder predicts the masked token using the unmasked tokens.

Our key insight is to interpret the signal reconstruction of MAE as a meta-learning task, where two disjoint unmasked and masked token sets are viewed as support and query sets to adapt and evaluate the Transformer meta-learner. To be specific, the Transformer encoder extracts the task knowledge through amortization of the support set, where this amortized latent adapts the Transformer decoder to predict the query set of the task. Formally, for a given data sample $\mathbf{x}$, we first divide the signal into two disjoint sets, namely the support set $\mathcal{S}_{\mathbf{x}}$ and the query set $\mathcal{Q}_{\mathbf{x}}$, by utilizing the tokenize operation $\texttt{tokenize}(\mathbf{x}) \coloneqq \{(m, \bar{\mathbf{x}}^{(m)})\}_{m=1}^M = \mathcal{S}_{\mathbf{x}} \cup \mathcal{Q}_{\mathbf{x}}$. Then, for a given Transformer encoder $f_\theta$ and decoder $g_\phi$, MAE minimizes the discrepancy between the predicted token and the corresponding masked token (i.e., the query sample) as:

$$\mathcal{L}_{\texttt{MAE}}(\theta, \phi; \mathcal{Q}_{\mathbf{x}}) \coloneqq \sum_{(q,\bar{\mathbf{x}}^{(q)}) \in \mathcal{Q}_{\mathbf{x}}} d\Big(\bar{\mathbf{x}}^{(q)}, g_\phi^{(q)}\big(\mathcal{Z}_{\mathbf{x}}\big)\Big) \ \text{ where } \ \mathcal{Z}_{\mathbf{x}} = f_\theta(\mathcal{S}_{\mathbf{x}}), \tag{1}$$

where $d(\cdot, \cdot)$ is a discrepancy function: $\ell_2$ norm for continuous (e.g., time-series, speech) and cross-entropy for discrete (e.g., token) datasets, respectively. Based on this interpretation, we improve the transfer ability of MAE (for modality-agnostic SSL) through a novel integration of two effective modality-agnostic meta-learning techniques to MAE.

### 2.2 MetaMAE: Improving Masked Auto-Encoder through Meta-Learning

We now describe our method, MetaMAE, which further improves the representation of MAE through a novel integration with advanced modality-agnostic meta-learning techniques. In a nutshell, Meta-MAE operates by further optimizing the amortized latent of the Transformer encoder using gradient-based meta-learning. Then we maximize the alignment between the optimized and the amortized latents via contrastive learning, to guide the Transformer encoder to improve the generalization.

---

**Algorithm 1** MetaMAE: Meta-Learning Modality-Agnostic Masked Auto-Encoder

---

**Require:** Unlabeled pretrain dataset $D_{\texttt{pretrain}}$, weight hyperparameter $\lambda$, Nearby-$\mathcal{S}$ ratio $r$,
        batch size $B$, learning rates $\alpha, \beta$.

---

1: Initialize $\theta, \phi, \psi$ using the standard initialization scheme.
2: **while** not done **do**
3:      Sample mini-batch $\mathcal{B} = \{\mathbf{x}_i\}_{i=1}^{B}$ from $D_{\texttt{pretrain}}$
4:      **for** $i = 1$ to $B$ **do**                                ▷ *Note: we use the batch computation.*
5:          Sample Support set $\mathcal{S}_{\mathbf{x}_i}$ and Query set $\mathcal{Q}_{\mathbf{x}_i}$ from $\mathbf{x}_i$
6:          $\mathcal{Z}_{\mathbf{x}_i} = f_\theta(\mathcal{S}_{\mathbf{x}_i})$                        ▷ Amortization through Transformer encoder.
7:          Sample $\mathcal{N}(\mathcal{S}_{\mathbf{x}_i}; r)$ where $|\mathcal{N}(\mathcal{S}_{\mathbf{x}_i}; r)| = r \times |\mathcal{Q}_{\mathbf{x}_i}|$      ▷ Sample the Nearby-$\mathcal{S}$ tokens.
8:          $\mathcal{Z}_{\mathbf{x}_i}^* \leftarrow \mathcal{Z}_{\mathbf{x}_i} - \alpha \nabla_{\mathcal{Z}_{\mathbf{x}_i}} \mathcal{L}_{\texttt{MAE}}(\theta, \phi; \tilde{\mathcal{S}}_{\mathbf{x}_i})$.
                 where $\tilde{\mathcal{S}}_{\mathbf{x}_i} = \mathcal{S}_{\mathbf{x}_i} \cup \mathcal{N}(\mathcal{S}_{\mathbf{x}_i}; r)$               ▷ Adapt the amortized latent.
9:          Compute MAE reconstruction loss $\mathcal{L}_{\texttt{grad}}^i$ with $\mathcal{Z}_{\mathbf{x}^i}^*$                ▷ Eq. (2)
10:         Compute task contrastive loss $\mathcal{L}_{\texttt{task-con}}^i$ with $\{\mathcal{Z}_{\mathbf{x}^i}\}_{i=1}^{B}$ and $\{\mathcal{Z}_{\mathbf{x}^i}^*\}_{i=1}^{B}$      ▷ Eq. (4)
11:         Compute MetaMAE loss $\mathcal{L}_{\texttt{MetaMAE}}^i$ with $\mathcal{L}_{\texttt{grad}}^i, \mathcal{L}_{\texttt{task-con}}^i$ and $\lambda$       ▷ Eq. (5)
12:      **end for**
13:      $\theta, \phi, \psi \leftarrow \theta, \phi, \psi - \frac{\beta}{B} \sum_{i=1}^{B} \mathcal{L}_{\texttt{MetaMAE}}^i$           ▷ Update the entire networks.
14: **end while**

---

**Latent adaptation via gradient-based meta-learning.** To further improve the generalization of MAE, we suggest utilizing the gradient-based meta-learning (i.e., model-agnostic meta-learning; MAML [22]) on the amortized latent space [74]. Specifically, we adapt the amortized latent of the support set to better reconstruct the support and the nearby tokens (of support tokens) by using the gradients of the decoder. Then, we utilize the optimized latent to condition the decoder to predict the query tokens. Here, our key idea is the use of nearby tokens when optimizing the latent, which turns out to be crucial for improved performance. Intuitively, optimizing such tokens induce an error correction on the latent, which eases the mask reconstruction (or prediction) task compared to the direct reconstruction, and thereby improves the task adaptation process [103].

Concretely, for a given support set $\mathcal{S}_{\mathbf{x}}$, we select the nearby tokens of support tokens from the query set $\mathcal{Q}_{\mathbf{x}}$, namely $\mathcal{N}(\mathcal{S}_{\mathbf{x}}; r) \subset \mathcal{Q}_{\mathbf{x}}$, such that the cardinality is $|\mathcal{N}(\mathcal{S}_{\mathbf{x}}; r)| = r \times |\mathcal{Q}_{\mathbf{x}}|$ with a ratio of $r > 0$. Then, we optimize the amortized latent $\mathcal{Z}_{\mathbf{x}}$ to better reconstruct the support and the nearby tokens $\tilde{\mathcal{S}}_{\mathbf{x}} := \mathcal{S}_{\mathbf{x}} \cup \mathcal{N}(\mathcal{S}_{\mathbf{x}}; r)$ using the decoder gradient, then condition the meta-learner, i.e., the Transformer decoder $g_\phi$, to predict the query set $\mathcal{Q}_{\mathbf{x}}$ as follows:

$$\mathcal{L}_{\texttt{grad}}(\mathbf{x}, \theta, \phi) := \sum_{(q, \bar{\mathbf{x}}^{(q)}) \in \mathcal{Q}_{\mathbf{x}}} d\Big(\bar{\mathbf{x}}^{(q)}, g_\phi^{(q)}\big(\mathcal{Z}_{\mathbf{x}}^*\big)\Big) \text{ where } \mathcal{Z}_{\mathbf{x}}^* = \mathcal{Z}_{\mathbf{x}} - \alpha \nabla_{\mathcal{Z}_{\mathbf{x}}} \mathcal{L}_{\texttt{MAE}}(\theta, \phi; \tilde{\mathcal{S}}_{\mathbf{x}}) \quad (2)$$

where $\alpha > 0$ is the step size for the adaptation. One can easily extend the latent optimization to obtain $\mathcal{Z}_{\mathbf{x}}^*$ with more than one gradient step where we found a single step adaptation is already quite effective yet showing computation efficiency compared to multiple iterations. Furthermore, we found that it is important to use the second-order gradients for the adaptation, i.e., backpropagation on the decoder adaptation gradient when optimizing the loss function, which enables the Transformer encoder to better amortize for the reconstruction task. Note that this gradient calculation on the decoder does not increase the computation too much, as using a smaller decoder size (compared to the encoder) is the key to the success of MAE [33].

**Task contrastive learning.** To further exploit the benefit of the gradient-based meta-learning, we suggest nudging the amortized latent to be as close as possible to the further optimized latent in Eq. (2). By doing so, the Transformer encoder is guided to better encode the reconstruction task knowledge as the optimized latent is further adapted with support and the nearby tokens. To effectively implement this concept, we utilize the idea of task contrastive learning [28, 60, 104]. Specifically, as both the optimized and amortized latents originate from the same task, we maximize the latent similarity within the same task while minimizing the similarity with other task latents.

Formally, let $\mathcal{Z}_{\mathbf{x}}$ and $\mathcal{Z}_{\mathbf{x}}^*$ be the amortized latent and optimized latent of the given input $\mathbf{x}$ from a mini-batch $\mathbf{x} \in \mathcal{B}$, respectively. We then use a non-linear projection network $h_\psi$ and the average set pooling of latent tokens to obtain task-specific representation $\mathbf{z}_{\mathbf{x}} = h_\psi\big(\text{pool}(\mathcal{Z}_{\mathbf{x}})\big)$ and $\mathbf{z}_{\mathbf{x}}^* = h_\psi\big(\text{pool}(\mathcal{Z}_{\mathbf{x}}^*)\big)$ for the task contrastive learning. For a given set of task-specific representations $\mathcal{T} = \bigcup_{\mathbf{x} \in \mathcal{B}}\{\mathbf{z}_{\mathbf{x}}, \mathbf{z}_{\mathbf{x}}^*\}$, the task contrastive objective is defined as follows:

$$\mathcal{L}_{\text{task-con}}(\mathbf{x}, \theta, \psi) := \frac{1}{2}\Big[ l_{\text{con}}(\mathbf{z}_{\mathbf{x}}; \mathbf{z}_{\mathbf{x}}^*, \mathcal{T}\setminus\{\mathbf{z}_{\mathbf{x}}^*\}) + l_{\text{con}}(\mathbf{z}_{\mathbf{x}}^*; \mathbf{z}_{\mathbf{x}}, \mathcal{T}\setminus\{\mathbf{z}_{\mathbf{x}}\})\Big] \qquad (3)$$

$$\text{where} \quad l_{\text{con}}(\mathbf{z}; \mathbf{z}^+, \{\mathbf{z}^-\}) := -\log \frac{\exp\big(\text{sim}(\mathbf{z}, \mathbf{z}^+)/\tau\big)}{\exp\big(\text{sim}(\mathbf{z}, \mathbf{z}^+)/\tau\big) + \sum_{\mathbf{z}^-}\exp\big(\text{sim}(\mathbf{z}, \mathbf{z}^-)/\tau\big)} \qquad (4)$$

where $\text{sim}(\mathbf{z}, \mathbf{z}') := \mathbf{z} \cdot \mathbf{z}'/\|\mathbf{z}\|\|\mathbf{z}'\|$ be the cosine similarity and $\tau > 0$ is the temperature hyper-parameter. From the perspective of contrastive representation learning, our task contrastive framework can be viewed as augmenting the positive pair. However, instead of using domain-specific inductive biases, we leverage gradient adaptation, thereby showing the possibilities of extending prior contrastive learning methods to modality-agnostic SSL frameworks.

**Overall meta-learning objective.** In the end, we derive a final training objective, $\mathcal{L}_{\text{MetaMAE}}$: a meta-learning objective combining the latent adaptation Eq. (2) and the task contrastive learning Eq. (4). For a given hyper-parameter $\lambda > 0$, the meta-objective of MetaMAE becomes:

$$\mathcal{L}_{\text{MetaMAE}}(\mathbf{x}, \theta, \phi, \psi) := \mathcal{L}_{\text{grad}}(\mathbf{x}, \theta, \phi) + \lambda\mathcal{L}_{\text{task-con}}(\mathbf{x}, \theta, \psi) \qquad (5)$$

## 3 Experiments

In this section, we demonstrate the effectiveness of the proposed framework by measuring the linear-evaluation performance under various datasets across modalities. We first describe our experimental setup (Section 3.1), and then we present the main experimental results (Section 3.2). We provide ablation studies regarding MetaMAE (Section 3.3).

### 3.1 Experimental Setup

We here briefly describe overall experimental setups. We provide further details of pretraining, evaluation, and hyperparameters in Appendix A.

**Datasets.** We select 8 sub-benchmarks from the DABS 2.0 benchmark [85], with categorizing the modalities for each sub-benchmark. We pretrain and transfer MetaMAE on the selected datasets:

- **Time-series** modality consists of datasets where the data is organized sequentially over time. In this paper, we use the PAMAP [71] dataset, which contains sensor signals from physical activity.

- **Tabular** modality refers to datasets where the data is structured in a table format, with rows (for instances) and columns (for attributes). We use the HIGGS [69] dataset from particle physics.

- **Multi-spectral (MS) Image** modality contains multi-channel 2D image datasets. We use the EuroSAT [34, 35] dataset, which consists of 13-channel satellite images.

- **Token** modality features datasets consisting of sequences of discrete units, similar to natural languages. We pretrain MetaMAE on both (a) the Genomics [72] dataset, subsequently transferring the learned model to the Genomics and Genomics-OOD datasets; and (b) the Pfam [20] dataset of proteins, followed by transfer learning to several tasks from the TAPE benchmarks [20], including Pfam, SCOP [23], Secondary Structure [43, 8], Stability [73], and Fluorescence [76].

- **Speech** modality includes 2D spectrograms of audio datasets. We pretrain MetaMAE on LibriSpeech [66], a large English audiobook corpus, and then transfer the model to datasets including LibriSpeech, Audio MNIST [6], Fluent Speech [57], Google Speech [96], and VoxCeleb1 [62].

- **RGB Image** modality comprises 3-channel 2D image datasets. We pretrain MetaMAE on (a) the ImageNet32 [17] dataset, which is scaled to $32 \times 32$, and transfer the pretrained model to datasets including CIFAR-10 [45], CUB [93], VGG Flowers [63], DTD [14], Traffic Sign [81], and Aircraft [58]; and (b) the WaferMap [98] dataset.

- **Vision-Language** modality comprises a combination of 3-channel 2D image and sequences of English text descriptions. We pretrain MetaMAE on MSCOCO [54], and then transfer the model to mismatched-caption detection [54] and the Visual Question Answering (i.e., VQA) tasks [1].

Table 1: In-domain linear evaluation performance across multiple modalities. We report F1-score (%) for WaferMap and the classification accuracy (%) for the rest. MS Image indicates the Multi-spectral image modality. *, and † denote the results from the DABS 1.0, and DABS 2.0 paper, respectively, where - of Capri results indicates that the pretraining loss divergence as described in [85].

| Modality | Time-series | Tabular | MS Image | Token | | Speech | RGB Image |
|---|---|---|---|---|---|---|---|
| Dataset | PAMAP2 | HIGGS | EuroSAT | Genom | Pfam | Libri | WaferMap |
| *Random initialization* | | | | | | | |
| Baseline | $69.8^\dagger$ | $54.8^\dagger$ | $62.3^\dagger$ | $37.2^\dagger$ | 30.1 | $17.1^*$ | $77.7^\dagger$ |
| *Self-supervised learning Framework* | | | | | | | |
| $e$-Mix | 80.1 | 65.7 | 87.4 | 40.5 | 31.3 | 60.2 | 92.6 |
| ShED | 85.2 | $68.0^\dagger$ | $61.5^\dagger$ | 33.6 | 54.7 | $34.8^*$ | $92.4^\dagger$ |
| Capri | - | - | $67.4^\dagger$ | $23.5^\dagger$ | 27.4 | 25.4 | $92.5^\dagger$ |
| MAE | $85.3^\dagger$ | $70.0^\dagger$ | $86.3^\dagger$ | 53.6 | 44.7 | 46.0 | $93.9^\dagger$ |
| **MetaMAE** | **89.3** | **71.5** | **88.5** | **69.4** | **62.3** | **79.8** | **95.5** |

Note that the transferred datasets can be in-domain (i.e., same dataset) or cross-domain (i.e., different dataset). The details of the benchmarks are described in Appendix C.

**Baselines.** For the main experiments, we compare MetaMAE's performance with existing modality-agnostic self-supervised learning methods suggested by DABS 1.0 [83], and 2.0 [85]:

- $e$**-Mix** is a generalized version of $i$-Mix [48], designed to consistently apply methods across both discrete and continuous domains by applying the mixup strategy in the embedding space.

- **ShED** is a generalized version of ELECTRA [15]. ShED constructs the pretext task, which involves predicting shuffled embeddings.

- **Capri** applys contrastive learning to the token level representation by randomly masking the token and treating different tokens as negative pairs.

- **MAE** aims to reconstruct the input. However, here, MAE employs a linear decoder for the continuous domain and no decoder for the discrete domain.

Additionally, we regard the randomly initialized encoder, referred to as the Baseline, as one of the baseline to check the effectiveness of self-supervised pretraining.

**Architectures.** Following [83, 85], we use 12 layers for the transformer encoder with the hidden size 256, and 8 attention heads. For the decoder, we fix the hidden size 128, and 4 attention heads. However, we choose an appropriate number of layers for the decoder to demonstrate the effect of the decoder for MAE. We also utilize different hyperparameters for each modality as other baselines, but we find that the hyperparameters can be shared across modalities (See Appendix B).

**Pretraining and transfer learning.** To evaluate our method, we pretrain each dataset 100K iterations and 100 epochs transfer learning, overall experiments by following [85]. We pretrain entire networks, i.e., encoder $f_\theta$, decoder $g_\theta$, and projection header $h_\theta$, but we utilize only the frozen encoder $f_\theta$ on transfer learning. When pretraining, the masking ratio can differ from the datasets. For the masking ratio hyper-parameter, we choose the best value among candidates suggested by the prior work [85].

## 3.2 Main Experiments

**In-domain linear evaluation.** We evaluate the pretrained representation on each in-domain downstream classification task. We report the performance of a linear classifier trained on top of the frozen features. The results in Table 1 demonstrate that our proposed method, MetaMAE, achieves state-of-the-art performance across the entire dataset. For instance, we obtain 16% accuracy gain ($53.6\% \to 69.4\%$) on Genomics. Moreover, we note that MAE has achieved moderate performance compared to other self-supervised learning (SSL) methods on these benchmarks, but MetaMAE demonstrates the ability to enhance MAE and outperform other SSL approaches. For example, MetaMAE achieves the best performance on Pfam ($44.7\% \to 62.3\%$) and LibriSpeech ($60.2\% \to 79.8\%$) with significant improvement, here is where MAE reported in [85] (i.e., MAE with linear decoder) was not the best among baselines.

Table 2: Cross-domain linear evaluation performance across multiple modalities. We report the Spearman correlation for Stability and Fluorescence datasets, and the classification accuracy (%) for the rest. * denote the results from the DABS 1.0 paper.

| Pretrain data | Transfer data | Baseline | SSL Framework | | | | |
| | | | e-Mix | ShED | Capri | MAE | **MetaMAE** |
|---|---|---|---|---|---|---|---|
| Genomics | Genomics-OOD | 8.6 | 9.7 | 7.3 | 5.5 | 22.2 | **37.2** |
| Pfam | SCOP | 8.0 | 5.7 | 10.7 | 2.0 | 7.9 | **11.8** |
| | Secondary | 52.4 | 53.7 | **67.6** | 49.5 | 62.5 | 65.9 |
| | Stability | 0.31 | 0.39 | **0.53** | 0.26 | 0.40 | **0.53** |
| | Fluorescence | 0.04 | 0.20 | 0.27 | 0.06 | 0.06 | **0.31** |
| LibriSpeech | Audio MNIST | 33.1* | 80.4* | 67.3* | 53.6 | 45.1 | **89.5** |
| | Fluent Loc | 62.1* | 60.9* | 60.2* | 59.8 | 61.7 | **66.7** |
| | Fluent Act | 26.2* | 29.9* | 30.5* | 28.3 | 26.8 | **38.4** |
| | Fluent Obj | 30.1* | 39.9* | 39.4* | 33.1 | 32.0 | **49.3** |
| | Google Speech | 4.9* | 19.2* | 20.7* | 13.7 | 9.5 | **46.8** |
| | VoxCeleb1 | 0.6* | 2.4* | 2.8* | 1.6 | 1.6 | **7.4** |
| ImageNet32 | CIFAR-10 | 24.2* | 39.4* | 39.6* | 48.7 | 46.0 | **59.2** |
| | CUB | 1.6* | 3.9* | 3.0* | 3.7 | 3.1 | **6.3** |
| | VGG Flowers | 9.0* | 26.0* | 13.0* | 18.6 | 22.2 | **36.3** |
| | DTD | 7.4* | 8.8* | 18.4* | 14.7 | 14.2 | **20.9** |
| | Traffic Sign | 14.3* | 65.1* | 27.5* | 28.0 | 32.0 | **67.1** |
| | Aircraft | 2.7* | 10.2* | 5.6* | 6.4 | 5.9 | **16.4** |

Table 3: Linear classification accuracy (%) pretrained on a vision-language dataset, MSCOCO.

| Pretrain data | Transfer data | Baseline | SSL Framework | | | | |
| | | | e-Mix | ShED | Capri | MAE | **MetaMAE** |
|---|---|---|---|---|---|---|---|
| MSCOCO | VQA | 53.4 | 57.6 | 53.1 | 52.9 | 54.2 | **69.7** |
| | Mismatched-caption | 49.8 | 50.1 | 50.6 | 49.6 | 49.3 | **70.5** |

**Cross-domain linear evaluation.** We evaluate our method on a diverse set of cross-domain downstream tasks including both classification and regression. We employ a linear classifier, or regressor trained on the frozen features as the in-domain setup. Table 2 shows that MetaMAE outperforms all the baselines across all the benchmarks consistently, except for one specific dataset. For example, we obtain 9% accuracy gain (80.4% → 89.5%) on the linear classification performance of transfer setup from LibriSpeech to Audio MNIST. It is important to note that cross-domain downstream tasks, due to their wider range of variations for each domain, are typically more challenging to consistently excel in compared to in-domain tasks. This significant performance improvement demonstrates the applicability of MetaMAE in various cross-domain transfer learning scenarios across the modalities.

**Multi-modal dataset evaluation.** One important future direction for the modality-agnostic SSL research community is to bind all modalities under a singular model [99, 108]. Here, we believe MetaMAE can be quite a promising method to tackle this problem, e.g., managing multiple modalities on a single model supplemented by domain-specific embedding modules. To this end, we verify the possibility of MetaMAE for tackling unified multi-modal self-supervised learning. As shown in Table 3, MetaMAE outperforms other modality-agnostic SSL methods on the vision-language tasks where we believe this multi-modal learning ability can help when unifying the modalities for SSL.

### 3.3 Ablation study

We perform an ablation study on six modalities: time-series (PAMAP2), tabular (HIGGS), speech (LibriSpeech), multi-spectral image (EuroSAT), and token (Pfam and Genomics). Throughout this section, we report the in-domain linear classification accuracy (%), unless otherwise specified.

**Component analysis.** In Table 4, we demonstrate the necessity of each component in MetaMAE by adding each component one by one: Deeper decoder $g_\phi$ with a Transformer architecture, latent optimization via gradient-based meta-learning, and the task contrastive loss $\mathcal{L}_{\texttt{task-con}}$. We first found

Table 4: Ablation study on each component of MetaMAE, namely the use of the decoder, latent adaptation using gradient-based meta-learning (Gradient-based), and task contrastive learning (Task contrast). We report the classification accuracy (%) across six different modalities.

| Decoder | Gradient-based | Task contrast | PAMAP2 | Genomics | EuroSAT | LibriSpeech | HIGGS | Pfam |
|---|---|---|---|---|---|---|---|---|
| ✗ | ✗ | ✗ | 85.3 | 53.6 | 86.3 | 33.3 | 70.0 | 44.7 |
| ✓ | ✗ | ✗ | 86.5 | 65.2 | 87.4 | 64.1 | 70.5 | 61.3 |
| ✓ | ✓ | ✗ | 88.3 | **69.4** | 87.4 | 64.5 | 71.1 | 61.3 |
| ✓ | ✓ | ✓ | **89.3** | **69.4** | **88.5** | **79.8** | **71.5** | **62.3** |

Table 5: Effect of the decoder size of MAE on the classification accuracy (%). We use three different datasets across modalities.

| decoder size | EuroSAT | Pfam | LibriSpeech |
|---|---|---|---|
| *prev. best* | **87.4** | 54.7 | 60.2 |
| 0 | 86.3 | 44.7 | 33.3 |
| 2 | 86.7 | **61.4** | 68.1 |
| 4 | **87.4** | 61.3 | 64.1 |
| 6 | 86.7 | **61.4** | **74.1** |

Table 6: Effect of the nearby token (i.e., Nearby-$\mathcal{S}$) selection ratio $r$ on the classification accuracy (%). We use three different datasets across modalities.

| $r$ ratio | PAMAP2 | HIGGS | Pfam |
|---|---|---|---|
| 0 | 87.5 | 71.1 | 62.0 |
| 0.1 | **89.3** | **71.5** | **62.3** |
| 0.5 | 88.2 | 70.8 | 62.0 |
| 1.0 | 84.2 | 70.1 | 62.1 |

that incorporating a deep decoder is a critical component in our framework, enabling domain-agnostic capabilities similar to the success of MAE on the image domain [33]. Thus, we here suggest that improving MAE for the domain-agnostic is quite a promising direction to explore.

In addition, Table 4 verifies the contribution of meta-learning schemes to the performance of Meta-MAE. We found that the gradient-based latent optimization rule, which includes the utilization of Nearby-$\mathcal{S}$, is more beneficial. We also confirm that task contrastive learning is a critical component in our framework like recent meta-learning frameworks [28, 60, 104]. Note that this task contrastive learning scheme is exclusively applicable in gradient-based approaches, emphasizing the significance of the gradient-based latent optimization method for MetaMAE.

**Importance of decoder size for MAE.** To verify the effect of decoder size for MAE, we evaluate the linear evaluation accuracy on datasets where the original MAE (i.e., no decoder) performed worse than other baselines. As Table 5 shows, we found that MAE can achieve the best performance compared to baselines by choosing the proper decoder size, yet there is room for enhancement as shown in Table 4. This result demonstrates the superiority of MAE for tackling modality-agnostic SSL problems, where we believe the development of MAE would be an important direction to investigate. In this respect, we believe MetaMAE will serve as an important baseline in this field.

**Nearby supports.** We further analyze the effect of $r$, i.e., the Nearby-$\mathcal{S}$ ratio. We conduct the experiment with $r \in \{0, 0.1, 0.5, 1.0\}$. We note that $r = 0$ indicates the gradient updates without any help of queries (i.e., direct reconstruction of $\mathcal{S}$), and $r = 1$ denotes the gradient updates with the entire queries near the $\mathcal{S}$. As shown in Table 6, this approach is found to be beneficial compared to the direct reconstruction, and the small ratio is suggested to be proper $r$, e.g., $r = 0.1$ is the best. This is because it effectively bridges the gap between the latent representation and the latents of masked tokens, thereby enhancing the encoding of knowledge required for the reconstruction task.

**Computational efficiency.** MetaMAE might be perceived as compute-inefficient when incorporating MAE due to the computational demands of second-order gradients; however, our findings suggest otherwise. Although MetaMAE increases the total training time of MAE by approximately 1.4 times (with the one-step adaptation), we have observed that it is much faster to achieve the best performance of MAE: in Figure 2, we compare the accuracy under the same training wall-clock time with MAE, e.g., 1.9 times faster on PAMAP2 dataset.

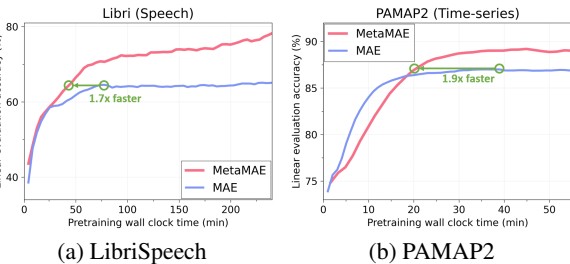

Figure 2: Computation efficiency comparison of MAE and MetaMAE. We report the pretraining wall clock time.

# 4 Related work

**Self-supervised learning (SSL).** SSL, i.e., learning without human supervision, recently has demonstrated substantial success across fields including, computer vision [32], natural language processing (NLP) [18], and speech recognition [2], frequently show better transferability and generalization ability compared to conventional pretraining methods, e.g., training on labeled datasets [11]. To learn such representation, SSL optimizes the loss on a pretext task that does not require any human labels. For instance, pioneer works for SSL proposed such tasks based on data reconstruction through auto-encoding [7] such as context prediction [19], and in-painting [67]. Later on, multiple SSL works found that utilizing the domain-specific inductive biases can effectively learn representations in a self-supervised manner, including, colorization [107], solving jigsaw puzzles [64], counting the number of objects [65], and rotation prediction [27], to name a few. More recently, contrastive representation learning has garnered significant attention in SSL [100, 32, 11]. This technique maximizes the similarities of similar (i.e., positive) pairs and minimizes the similarities of dissimilar (i.e., negative) pairs, rather than focusing on training an instance classifier. To generate such positive pairs, multiple works rely on domain-specific inductive biases such as data augmentations [47], i.e., different augmented views as a positive pair. In addition, recent advances have been made with the development of various architectural components: e.g., Siamese networks [44], self-distillation [32, 9], asymmetric architectures [29, 12], and utilization of Transformer architectures [9]. Despite the success of these strategies, most existing SSL frameworks rely heavily on domain-specific inductive biases, which limits their applicability to new modalities.

**Modality-agnostic SSL.** Recently, several streams of work have emerged focusing on the development of more generalized SSL methods, specifically modality-agnostic SSL. For example, DACL [90] and $i$-Mix [48] utilize the idea of mixup [105] to propose domain-agnostic contrastive learning, and $e$-Mix [83] generalizes the concept of $i$-Mix to be embedding-level instead of input-level. Capri [85], as a variant of CPC [88], contrasts the predicted representations from randomly masked tokens. [84] develops generative models to learn data-dependent distortions for contrast. Instead of contrastive learning, ShED [83] (a generalized version of ELECTRA [15]) constructs the pretext task of replacing token detection with a masking strategy. DABS 2.0 [85] proposes a method to generalize MAE [33] to be modality-agnostic. In their approach, however, decoders are not utilized for discrete domains like BERT [18], while only a linear decoder is employed for continuous domains. In this paper, we suggest an effective modality-agnostic latent optimization for learning representations by interpreting masked prediction for MAE [33] in a novel manner.

**Masked Auto-Encoder (MAE).** MAE [33], i.e., predicting the masked parts with a given unmasked parts, has been extended to multiple applications [109, 37] across various domains [18, 39]. Among them, the recent combination of MAE with Transformer architecture [18, 33, 5] has shown promise in tackling SSL scenarios. For instance, BERT [18] utilized MAE for natural language processing (NLP) tasks, incorporating a linear layer into its architecture. Furthermore, multiple variants of MAE show impressive performance in various domains, by suggesting modality-agnostic SSL [3, 4], architecture-agnostic SSL [97, 51], multi-modal pretraining [95], and generative pretraining frameworks [21, 52]. In this paper, we focus on improving the most basic form of mask-modeling (i.e., MAE) for constructing a modality-agnostic SSL framework which remains under-explored, despite its potential significance, through the lens of meta-learning. It is worth noting that our interpretation of viewing MAE as a meta-learning framework can be applied to any other masked-modeling-based SSL frameworks where we believe combining our meta-learning regularization to such SSL methods would be an interesting direction to explore.

**Meta-learning.** Meta-learning [86], i.e., learning to learn by extracting common knowledge over a task distribution, has emerged as a popular paradigm for enabling systems to adapt to new tasks in a sample-efficient way. Under various applications across domains (e.g., computer vision [78], natural language processing [30], and robotics [101]), there have been significant efforts to design a variety of meta-learning schemes, including gradient-based [22, 53] and amortization-based approaches [75, 61] such as metric-based [92, 79], and neural processes [25, 24, 41, 87]. Typically, recent works have combined gradient-based meta-learning (or iterative functional update) with amortization-based schemes to enhance adaptation performance [74, 103]. Furthermore, there have been varieties of amortization-based schemes (such as neural processes) that utilize the recent success of contrastive learning into meta-learning, i.e., task contrastive learning [28, 60]. In this paper, we interpret MAE as an amortization-based meta-learning, which is further enhanced via the benefit of model-agnosticism of gradient-based meta-learning and task contrastive learning.

# 5 Discussion and conclusion

In this paper, we tackle modality-agnostic self-supervised learning (SSL), an important problem of SSL that consists of multiple real-world applications. To this end, we explore the possibilities of the Masked Auto-Encoder (MAE) in tackling modality-agnostic SSL which is quite under-explored, despite its potential. We propose MetaMAE, a novel and effective SSL framework that enhances MAE with meta-learning. Our key idea is to interpret mask reconstruction task of MAE as a meta-learning task, which allows us to treat MAE as a meta-learning framework. Based on this novel interpretation, we suggest a unique integration with advanced modality-agnostic meta-learning methods to improve the generalization of MAE. Our experiments demonstrate that MetaMAE significantly improves the performance of modality-agnostic SSL approaches across a diverse range of modalities.

**Limitations and future work.** While MetaMAE becomes a state-of-the-art approach for modality-agnostic SSL problems, it still inherits a general limitation of the MAE, namely the modality-specific masking ratio, i.e., the masking ratio may differ across modalities. This is due to our shared design elements with MAE, which include masking, encoding, and decoding. Recent works propose design choices for the masking scheme [50, 94], including automation, where incorporating these ideas into MetaMAE would be an intriguing future research direction, potentially enhancing our approach to be an even more effective modality-agnostic SSL framework.

**Potential negative impacts.** SSL often requires a large computation and a large network capacity, therefore raising environmental concerns, e.g., carbon generation [77]. As MetaMAE is built upon the SSL method (i.e., MAE), practitioners may need to consider some computation for successful training. To address this issue, efficient training methods [80, 40], distilling knowledge to a smaller network [36], or network sparsity schemes [31, 46] would be required to ameliorate such problems.

## Acknowledgements

We thank Kyungmin Lee for providing helpful feedbacks and suggestions in preparing an earlier version of the manuscript. We also thank Sang Keun Choe for technical suggestions on the PyTorch implementation of meta-learning. This work was supported by Institute of Information & communications Technology Planning & Evaluation (IITP) grant funded by the Korea government (MSIT) (No.2019-0-00075, Artificial Intelligence Graduate School Program (KAIST); No.2021-0-02068, Artificial Intelligence Innovation Hub; No.2022-0-00959, Few-shot Learning of Causal Inference in Vision and Language for Decision Making) and Samsung Electronics Co., Ltd (IO201211-08107-01).

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

# A   Implementation details

In this section, we provide the implementation details of MetaMAE, including architectures and hyperparameters for MetaMAE when pretraining and evaluation.

**Architectural details.** We summarize our architectures in Table 7, with the hyperparameter notation referred from [89]. We use token embedding for encoder inputs, and apply positional embedding to both encoder and decoder inputs, as suggested by [89]. Specifically, token embedding separates the input data into fixed-size tokens, while positional embedding uses a fixed, absolute position represented by a combination of sine and cosine functions. We describe the token size for each specific dataset in Appendix C.

Table 7: A Pytorch-like architecture description of MetaMAE. $n \in \{2, 4, 6\}$, $p \in \{0, 0.1\}$ are the hyperparameters.

| Component | Layer descriptions |
|---|---|
| Encoder $f_\theta$ | TransformerBlock($d_{\text{model}} = 256, d_{\text{ff}} = 512, h = 8, P_{\text{drop}} = p$, GELU, LayerNorm=True) $\times$ 12 |
| Decoder $g_\phi$ | TransformerBlock($d_{\text{model}} = 128, d_{\text{ff}} = 256, h = 4, P_{\text{drop}} = 0$, GELU, LayerNorm=True) $\times$ $n$ |
| Projector $h_\psi$ | Linear($256, 1028$), BatchNorm1d($1028$), Linear($1028, 128$) |

**Pretraining details.** We summarize our selected hyperparameters for pretraining each dataset in Table 8. Following [85], we pretrain MetaMAE for 100k iterations utilizing the AdamW optimizer [56] with both a learning rate and weight decay set at 1e-4. The batch size for pretraining and the strategy for selecting the mask ratio are detailed in [83, 85]. For the MetaMAE-specific hyperparameters, we observe that a certain set of hyperparameters can generally work across modalities, e.g., ($\alpha$, $\lambda$, decoder depth) = (0.5, 0.1, 4) (see Table 9 in Appendix B), or can be shared within each modality, e.g., $P_{\text{drop}} = 0$ for Token modality (see Table 10 in Appendix B). Nevertheless, we recommend modality-specific values for optimal performance (refer to Appendix B for hyperparameter sensitivity details). We set the temperature term for the contrastive loss $\tau = 0.5$ and the Nearby-$\mathcal{S}$ ratio $r = 0.1$. For latent adaptation, the latent representation undergoes a single-step update with the update magnitude denoted by $\alpha$.

Table 8: Hyperparameters of MetaMAE for pretrain datasets.

| Modality | Time-series | Tabular | MS Image | Token | | Speech | RGB Image | | Vision-Language |
|---|---|---|---|---|---|---|---|---|---|
| **Dataset** | PAMAP2 | HIGGS | EuroSAT | Genom | Pfam | Libri | WaferMap | ImageNet32 | MSCOCO |
| *MetaMAE-specific hyperparameters* | | | | | | | | | |
| $\alpha$ | 0.5 | 1.0 | 0.1 | 0.1 | 0.1 | 0.1 | 0.1 | 0.1 | 0.5 |
| $\lambda$ | 1.0 | 1.0 | 1.0 | 0.01 | 1.0 | 1.0 | 0.1 | 0.1 | 0.1 |
| decoder depth | 4 | 6 | 4 | 2 | 4 | 4 | 6 | 4 | 2 |
| $P_{\text{drop}}$ | 0.1 | 0.1 | 0 | 0 | 0 | 0 | 0 | 0 | 0 |
| *Hyperparameters from DABS benchmarks* | | | | | | | | | |
| mask ratio | 0.85 | 0.50 | 0.85 | 0.50 | 0.15 | 0.85 | 0.15 | 0.85 | 0.5 |
| batch size | 256 | 256 | 64 | 32 | 128 | 64 | 128 | 64 | 64 |

We note that to scale up experiments, it is essential to facilitate distributed parallelism by using libraries such as BETTY [13] when utilizing PyTorch for meta-learning.

**Evaluation details.** In line with [85], we freeze the pretrained model and train either a linear classifier or a regressor for 100 epochs during the linear evaluation phase. We use the Adam optimizer [42] with both the learning rate and weight decay set as 1e-4. The batch size for this linear evaluation is set as described in [83, 85].

# B    Analysis on hyperparameter sensitivity

We here provide additional experiments on hyperparameters. This includes sharing various hyperparameters across modalities and conducting ablation studies with varying hyperparameters: $\alpha$, $\lambda$, decoder depth, $P_{\text{drop}}$, and the latent adaptation step size.

**Sharing hyperparameters across modalities.** As demonstrated in Table 9, MetaMAE shows robust performance regardless of the hyperparater selection. Notably, two or three major hyperparameters can be shared across all modalities, still outperforming prior methods. Furthermore, Table 10 indicates the resilience of MetaMAE's pretraining hyperparameters, especially at the intra-modality level.

Table 9: Linear evaluation performance (%) across modalities. Sharing 2 and 3 HPs denotes MetaMAE with additionally sharing more hyperparameters among the non-shared hyperparameters in Table 8 in Appendix A, which are $(\alpha, \lambda) = (0.5, 0.1)$ and $(\alpha, \lambda, \text{decoder depth}) = (0.5, 0.1, 4)$, respectively. HP denotes hyperparameter.

| Modality | Time-series | Tabular | MS Image | Token | | Speech | RGB Image |
|---|---|---|---|---|---|---|---|
| Dataset | PAMAP2 | HIGGS | EuroSAT | Genom | Pfam | Libri | WaferMap |
| $e$-Mix | 80.1 | 65.7 | 87.4 | 40.5 | 31.3 | 60.2 | 92.6 |
| ShED | 85.2 | 68.0 | 61.5 | 33.6 | 54.7 | 34.8 | 92.4 |
| Capri | - | - | 67.4 | 23.5 | 27.4 | 25.4 | 92.5 |
| MAE | 85.3 | 70.0 | 86.3 | 53.6 | 44.7 | 46.0 | 93.9 |
| **MetaMAE (sharing 3 HPs)** | 89.1 | 71.0 | **88.5** | 55.4 | 62.2 | 77.1 | 95.4 |
| **MetaMAE (sharing 2 HPs)** | 89.1 | 71.1 | **88.5** | 66.7 | 62.2 | 77.1 | 95.4 |
| **MetaMAE (reported)** | **89.3** | **71.5** | **88.5** | **69.4** | **62.3** | **79.8** | **95.5** |

Table 10: Linear evaluation performance (%) with sharing all hyperparameters, except mask ratio, intra-modality level. For sharing HPs in intra-modal, we use $(\alpha, \lambda, \text{decoder depth}, P_{\text{drop}}) = (0.1, 0.01, 2, 0)$ and $(0.1, 0.1, 4, 0)$ for Token and RGB Image modalities, respectively. HP denotes hyperparameter.

| Modality | Token | | RGB Image | |
|---|---|---|---|---|
| Dataset | Genom | Pfam | WaferMap | ImageNet32 $\rightarrow$ CIFAR10 |
| **MetaMAE (sharing HPs in intra-modality)** | 69.4 | 61.5 | 95.5 | 59.2 |
| **MetaMAE (reported)** | 69.4 | 62.3 | 95.5 | 59.2 |

**Further ablation studies with varying hyperparameters.** Table 11, 12, 13, and 14 show the sensitivity of hyperparameters on the PAMAP2 and WaferMap datasets. We observe that MetaMAE performs well even with non-optimal hyperparameters, except for the decoder depth and $P_{\text{drop}}$, but we suggest finding better hyperparameters specific to each domain (e.g., $\lambda = 0.1$ for WaferMap). Regarding the decoder depth, we find that each modality requires an appropriate value, but generally, MetaMAE performs well with a decoder depth of 4. In Table 15, we observe that single-step adaptation effectively achieves good performance, and in some cases, even outperforms multiple-step adaptation due to the risk of overly decoder-specific support representation.

Table 11: Sensitivity of $\alpha$ on PAMAP2 and WaferMap.

| $\alpha$ | PAMAP2 | WaferMap |
|---|---|---|
| 0.1 | 86.2 | **95.5** |
| 0.5 | **89.3** | 95.4 |
| 1.0 | 89.1 | 95.2 |

Table 12: Sensitivity of $\lambda$ on PAMAP2 and WaferMap.

| $\lambda$ | PAMAP2 | WaferMap |
|---|---|---|
| 0.01 | 88.6 | 95.2 |
| 0.1 | **89.1** | **95.5** |
| 1.0 | **89.3** | 93.6 |

Table 13: Sensitivity of decoder depth on PAMAP2 and WaferMap.

| depth | PAMAP2 | WaferMap |
|---|---|---|
| 2 | 84.9 | 94.2 |
| 4 | **89.3** | **95.5** |
| 6 | 86.2 | **95.5** |

Table 14: Sensitivity of $P_{\text{drop}}$ on PAMAP2 and WaferMap.

| $P_{\text{drop}}$ | PAMAP2 | WaferMap |
|---|---|---|
| 0 | 79.4 | **95.5** |
| 0.1 | **89.3** | 94.7 |

Table 15: Sensitivity of latent adaptation step size on PAMAP2 and WaferMap.

| step size | PAMAP2 | WaferMap |
|---|---|---|
| 1 | 89.3 | **95.5** |
| 5 | **89.6** | 94.9 |

# C Dataset details

We provide a summary of the considered datasets from the DABS benchmarks [83, 85] in Table 16. Note that we use the dataset split described in [83, 85].

Table 16: Datasets considered for pretraining and linear evaluation in our experiments. "MS Image" denotes the Multi-spectral image modality. For Phase, "P" denotes pretraining and "F" denotes fine-tuning.

| Modality | Dataset | # of classes | Input shape | Token shape | Phase | Batch size |
|---|---|---|---|---|---|---|
| Time-series | PAMAP2 [71] | 12 | $52 \times 320$ | 5 | P & F | 256 & 256 |
| Tabular | HIGGS [69] | 2 | 28 | 1 | P & F | 256 & 256 |
| MS Image | EuroSAT [35, 34] | 10 | $13 \times 64 \times 64$ | $8 \times 8$ | P & F | 64 & 64 |
| Token | Genomics [72] | 10 | $4 \times 250$ | 1 | P & F | 32 & 64 |
| | Genomics-OOD [72] | 60 | $4 \times 250$ | 1 | F | 32 |
| | Pfam [20] | 623 | $26 \times 128$ | 1 | P & F | 128 & 128 |
| | SCOP [23] | 1195 | $26 \times 128$ | 1 | F | 128 |
| | Secondary Structure [43, 8] | 4 | $26 \times 128$ | 1 | F | 128 |
| | Stability [73] | - | $26 \times 128$ | 1 | F | 128 |
| | Fluorescence [76] | - | $26 \times 128$ | 1 | F | 128 |
| Speech | LibriSpeech [66] | 40 | $1 \times 224 \times 224$ | $16 \times 16$ | P & F | 64 & 64 |
| | Audio MNIST [6] | 10 | $1 \times 224 \times 224$ | $16 \times 16$ | F | 64 |
| | Fluent Locations [57] | 4 | $1 \times 224 \times 224$ | $16 \times 16$ | F | 64 |
| | Fluent Actions [57] | 6 | $1 \times 224 \times 224$ | $16 \times 16$ | F | 64 |
| | Fluent Objects [57] | 14 | $1 \times 224 \times 224$ | $16 \times 16$ | F | 64 |
| | Google Speech [96] | 36 | $1 \times 224 \times 224$ | $16 \times 16$ | F | 64 |
| | VoxCeleb1 [62] | 1251 | $1 \times 224 \times 224$ | $16 \times 16$ | F | 64 |
| RGB Image | waferMap [98] | 9 | $3 \times 32 \times 32$ | $4 \times 4$ | P & F | 128 & 128 |
| | ImageNet-32 [17] | 1000 | $3 \times 32 \times 32$ | $4 \times 4$ | P | 64 |
| | CIFAR-10 [45] | 10 | $3 \times 32 \times 32$ | $4 \times 4$ | F | 64 |
| | CUB [93] | 200 | $3 \times 32 \times 32$ | $4 \times 4$ | F | 64 |
| | VGG Flowers [63] | 102 | $3 \times 32 \times 32$ | $4 \times 4$ | F | 64 |
| | DTD [14] | 47 | $3 \times 32 \times 32$ | $4 \times 4$ | F | 64 |
| | Traffic Sign [81] | 43 | $3 \times 32 \times 32$ | $4 \times 4$ | F | 64 |
| | AirCraft [58] | 102 | $3 \times 32 \times 32$ | $4 \times 4$ | F | 64 |
| Vision-Language | MSCOCO [54] | 80 | $(3 \times 224 \times 224, 30552 \times 32)$ | $(16 \times 16, 1)$ | P | 64 |
| | VQA [1] | 2 | $(3 \times 224 \times 224, 30552 \times 32)$ | $(16 \times 16, 1)$ | F | 64 |
| | Mismatched-caption [54] | 2 | $(3 \times 224 \times 224, 30552 \times 32)$ | $(16 \times 16, 1)$ | F | 64 |

