# OpenReview forum: "Modality-Agnostic Self-Supervised Learning with Meta-Learned Masked Auto-Encoder"
_NeurIPS.cc/2023/Conference — NeurIPS 2023 poster_

### Official Review · Reviewer_Pe9M · 2023-07-06

**Soundness:** 3 good
**Presentation:** 3 good
**Contribution:** 3 good
**Rating:** 5
**Confidence:** 3

**Summary:**

This paper presents Meta-learned Masked Auto-Encoder (MetaMAE), a novel modality-agnostic self-supervised learning (SSL) framework that leverages meta-learning to improve the transfer abilities of Masked Auto-Encoder (MAE). The authors reinterpret the mask reconstruction task of MAE as a meta-learning task and propose the integration of two advanced meta-learning techniques: gradient-based meta-learning and task contrastive learning. MetaMAE is evaluated on various data modalities from modality-agnostic SSL benchmarks, demonstrating significant improvements over previous modality-agnostic SSL methods in linear evaluation. The proposed approach also shows improved transferability on cross-domain datasets.

**Strengths:**

1. The paper presents an innovative approach to modality-agnostic SSL by leveraging meta-learning and reinterpreting mask reconstruction in MAE as a meta-learning task.

2. This paper is well written.

3. MetaMAE consistently outperforms previous modality-agnostic SSL methods in linear evaluation, indicating its effectiveness.

4. Contrastive learning in MetaMAE can improve performance.

**Weaknesses:**

1. There are significant differences in pretraining and fine-tuning hyperparameters for various downstream tasks, such as masking ratio, batch size, and decoder depth, as shown in Appendix Table 2.

2. The paper lacks innovation.

3. In the ablation experiments, it can be observed that the proposed Gradient-based and Task contrast methods have minimal impact on most tasks.


**Questions:**

N/A

**Limitations:**

addressed

---

> ### Author Rebuttal · Authors · 2023-08-08
>
> Dear Reviewer Pe9M,
>
> We sincerely appreciate your efforts and insightful comments to improve the manuscript. We address your comment below.
>
> ---
>
> **[Q1] Significant difference in hyper-parameters for various downstream tasks, e.g., masking ratio, batch size, and decoder depth.** \
> **[A1]** For your concern, we first clarify that except for the additional hyper-parameters introduced by MetaMAE, we follow the same hyper-parameter selection strategy from the prior work, DABS [1] (i.e., the paper introducing the modality-agnostic SSL benchmark). Specifically, we use the same hyper-parameters for the linear evaluation (i.e., fine-tuning; see Lines 17-21 in Appendix) and also use the same hyper-parameter selection for pre-training (e.g., batch size, optimization hyper-parameters, and choose the best-performing mask ratio from candidates in [1] by following [1]).
>
> Regarding additional hyper-parameters of MetaMAE, it is worth noting that it is a common consensus that the hyper-parameter differs across modalities, e.g., BERT (for text) uses a mask ratio of 0.15, and MAE (for image) uses 0.75. Nevertheless, as reported in Table 3 of Appendix A, even sharing most hyper-parameters across *all modalities*, MetaMAE consistently outperforms previous state-of-the-art results across all modalities. Furthermore, we run *additional* hyper-parameter ablation studies (see Table 2 in the attached pdf), which further support that the choice of pre-training hyper-parameters of MetaMAE is particularly robust at the intra-modality level. For example, for a Token modality, shared hyper-parameters $(\alpha, \lambda, \text{decoder depth}, P_\text{drop}) = (0.1, 0.01, 2, 0)$ can achieve 69.4\% on Genom and 61.5\% on Pfam, which are very close (or identical) to the reported performance of MetaMAE (69.4\%, and 62.3\%, respectively) in our original submission. These results could be useful observations in practice when pretraining a certain dataset of the same modality we experimented with.
>
> Finally, we also agree with the reviewer that developing a unified way of selecting hyper-parameters across modalities is an important and interesting direction. In this perspective, we believe MetaMAE can contribute as a strong baseline for future research in this direction, showing a robust performance regardless of the hyper-parameter selection (see Table 3 in the attached pdf); two major hyper-parameters ($\alpha, \lambda$) can be additionally shared across all modalities while it shows competitive performance with the MetaMAE in our original submission. We will include all results in the final manuscript.
>
> ---
>
> **[Q2] The paper lacks innovation.** \
> **[A2]** We politely yet strongly disagree with this claim. We believe our approach is novel and innovative as highlighted by Reviewers GPRA, 2wfj, and Pe9M, where the key novelty is to interpret MAE as a meta-learning framework. This novel interpretation enables us to incorporate advanced meta-learning schemes which lead to an enhanced performance.
>
> Furthermore, we believe the experimental results also reflect our innovation. Typically, as SSL is becoming more important in multiple real-world industrial domains [1], the superiority of MetaMAE will be highly impactful for various applications across modalities. Tackling modality-agnostic SSL will be even more crucial as new types of modality datasets (or industrial applications) increase, where we believe MetaMAE will serve as an important future baseline in this field.
>
> [1] Tamkin et al., DABS: A Domain-agnostic Benchmark for Self-supervised Learning, NeurIPS 2021 Track on Datasets and Benchmarks
>
> ---
>
> **[Q3] In the ablation experiments, Gradient-based and Task contrast methods have minimal impact on most tasks.** \
> **[A3]** We politely disagree with your claim. Note the cases with maximal or significant improvements in Table 4 in the attached pdf. For instance, gradient adaptation increases by 1.7\% on PAMAP2, and task contrastive learning improves by 15.3\% on Librispeech. Moreover, to further address your concern, we conduct an additional ablation study on Genomics (See the same table above), where gradient adaptation improves by 4.2\%. This result shows that our components contribute considerably large performance gains in some cases.
>
> Furthermore, we would like to note that the gradient/task contrastive consistently improves the performance across multiple modalities as we reported ablation experiments on a lot of modalities in Table 3 of the main paper. We remark that consistent improvement is an important aspect when targeting multi-modal domains (e.g., Table 1 in the attached pdf for your information), as it suggests that the approach generalizes well across different domains and is not tailored to optimize only specific ones.

---

> > ### Comment · Reviewer_Pe9M · 2023-08-15
> > **Re: Rebuttal by Authors**
> >
> > Thanks for your rebuttal for addressing my comments. I have raised the rating.

---

> > > ### Author Response · Authors · 2023-08-15
> > > **Thank you for the response**
> > >
> > > Dear reviewer Pe9M,
> > >
> > > Thank you for letting us know! We are more than happy to hear that our rebuttal addressed your questions well. \
> > > If you have any further questions or suggestions, please do not hesitate to let us know.
> > >
> > > Thank you very much,\
> > > Authors

---

### Official Review · Reviewer_9KfC · 2023-07-07

**Soundness:** 3 good
**Presentation:** 2 fair
**Contribution:** 2 fair
**Rating:** 5
**Confidence:** 3

**Summary:**

The authors of this paper address modality-agnostic self-supervised learning (SSL), a significant yet under-researched problem with various real-world applications. They explore the potential of the Masked Auto-Encoder (MAE) in this context and propose SSL framework, MetaMAE, which enhances MAE with meta-learning. They reinterpret the mask reconstruction task of MAE as a meta-learning task, which in turn allows them to apply modality-agnostic meta-learning methods to improve MAE's generalization. Experimental results reveal that MetaMAE improve the performance of modality-agnostic SSL approaches across multiple modalities.

**Strengths:**

This paper views Masked Auto-Encoder (MAE) from a meta-learning perspective, which is interesting.
The paper is well-structured, allowing for easy comprehension of the information flow.
The explanation on how support and query sets corresponds to tokens is clear.
The experiment section is solid. The paper provides a comprehensive evaluation of the proposed method through thorough experiments and comparisons with existing models.
The approach shows great generalizability on cross-domain linear evaluation task.

**Weaknesses:**

The experiment setup for meta-learning based gradient update could benefit from more explanation. Like how is the support and query set constructed during training for gradient updates.

Meta-learning usually requires more compute due to the computation of second order gradients, and discussion on computational overhead could be necessary.

Despite the variety of tasks for evaluation, most of the datasets used for evaluation is of small-scale.

One important ablation study missing to validate the effectiveness of the proposed overall framework. (See question below)

**Questions:**

In ablation study section, Tab.3, why not use only task contrast without gradient-based component to see how much of the gain comes from contrastive learning itself?

Also, how to balance the meta-learning gradient update and contrastive learning update? Would they influence each other? Is the hyper-parameter lambda grid searched?

**Limitations:**

Masking scheme limitation on MAE is discussed.

---

> ### Author Rebuttal · Authors · 2023-08-08
>
> Dear Reviewer 9KfC,
>
> We sincerely thank you for your helpful feedback and insightful comments. We address your comments and questions below.
>
> ---
>
> **[Q1] Explanation of the experimental setup for meta-learning-based gradient update, e.g., how the support and query set is constructed during training.** \
> **[A1]** Thank you for the suggestion to enhance our manuscript. During pretraining, we randomly split the tokenized data into two disjoint sets, denoted as the support and query sets. Note that support and query sets can be interpreted as unmasked and masked tokens, respectively, in the MAE context. For linear evaluation, we consider only the support set to obtain representations for both train and test samples, following the standard self-supervised learning setup. We will include the explanations for the experimental setup in the final manuscript.
>
> ---
>
> **[Q2] Discussion of computational overhead due to the computation of second-order gradients.** \
> **[A2]** Thank you for your insightful comment. It is true that second-order gradients require more computation, but we found that MetaMAE can be somewhat efficiently implemented by using one-step adaptation. Also, using a small decoder size compared to the encoder size is the key to MAE, thereby the gradient computation is quite efficient.
>
> Furthermore, we found that while MetaMAE increases the total training time of MAE by roughly 1.4 times, we have observed that it is much faster to achieve the best performance of MAE: in Figure 1 in the attached pdf, we compare the accuracy under the same training wall-clock time with MAE; for example 1.7 times faster on LibriSpeech. Thank you for your comment and will incorporate this result in the final manuscript.
>
> ---
>
> **[Q3] Most of the datasets used for evaluation are small-scale.** \
> **[A3]** We follow the evaluation protocol of the standard modality-agnostic SSL benchmark (i.e., DABS), which also consists of considerably large-scale datasets (e.g., LibriSpeech, a large-scale audio dataset, which is used for pretraining and evaluating well-known audio SSLs [1,2]), thereby showing the scalability of MetaMAE. Nonetheless, we agree that the size of some datasets in the DABS can be relatively small than other SSL benchmarks (as they are collected from practical applications), and evaluating more large-scale experiments can further demonstrate the scalability of MetaMAE. To this end, we refer our experimental results on MSCOCO (consisting of 328K image caption pairs), which was done to address the discussion on the potential advantages of MetaMAE on multimodal SSL. We found that MetaMAE outperforms other modality-agnostic SSL approaches for both the two vision-language downstream tasks (see Table 1 in the attached pdf; e.g., 57.6\% -> 69.7\% on the VQA task). In the final manuscript, we will include the above result and highlight the promise of MetaMAE for large-scale SSL training.
>
> [1] Baevski et al., wav2vec 2.0: A Framework for Self-supervised Learning of Speech Representations, NeurIPS 2020 \
> [2] Hsu et al., HuBERT: Self-supervised Speech Representation learning by Masked Prediction of Hidden Units, ACM 2021
>
> ---
>
> **[Q4] Missing ablation study: Why not use only task contrast without a gradient-based component?** \
> **[A4]** We would like to clarify that the ablation study you suggested is not possible as the gradient-based component is essential for task contrast. To perform task contrastive learning, positive pairs must be defined, and in our framework, they were defined as pairs of amortized and adapted latents. Remark that the adapted latents are constructed from the gradient-based component, which means that we can exploit task contrastive learning thanks to the gradient-based component.
>
> ---
>
> **[Q5] How to balance the meta-learning gradient update and contrastive learning update? Would they influence each other? Is the hyper-parameter lambda grid searched?** \
> **[A5]** Thank you for the question. We employ a log-scaled grid search for the hyper-parameter $\lambda$ to regulate only the contrastive loss. However, the two loss terms can be influenced by each other as larger $\lambda$ can emphasize more similar representations for positive pairs. This makes the model learn toward smaller gradient differences between the amortized and adapted latents. We would like to note that $\lambda = 0.1$ consistently outperforms the previous results. This indicates that $\lambda$ is not sensitive to the specific modalities, yet finding specific values for each modality is helpful for achieving optimal performance.

---

> > ### Comment · Reviewer_9KfC · 2023-08-19
> >
> > Thank you for your response. After reading I keep my original score and recommend acceptance.

---

> > > ### Author Response · Authors · 2023-08-19
> > > **Thank you for the response**
> > >
> > > Dear reviewer 9KfC,
> > >
> > > Thank you for letting us know! We are happy to hear that our rebuttal addressed your questions well. \
> > > If you have any further questions or suggestions, please do not hesitate to let us know.
> > >
> > > Thank you very much, \
> > > Authors

---

### Official Review · Reviewer_b3vS · 2023-07-10

**Soundness:** 3 good
**Presentation:** 3 good
**Contribution:** 3 good
**Rating:** 7
**Confidence:** 2

**Summary:**

This paper studies self-supervised learning (SSL) for multiple modalities (e.g. tabular data, images, text, speech). Central to this paper are Masked Auto-Encoders (MAEs), a popular SSL training paradigm where models are tasked to reconstruct inputs from randomly masked subsets. The authors interpret this reconstruction task as a meta-learning task, and based on this interpretation, propose a new method to improve the performance of MAE, called MetaMAE. The proposed method provides solid improvements over previous work on several datasets and modalities from the DABS benchmark.

**Strengths:**

- The results presented in the paper are quite strong, surpassing previous results often by a large margin. As such, I believe this paper presents a solid advance over existing literature, and would be of interest to most researchers working on SSL.
- The paper is clear and well written.
- The experiments are solid and extensive, and there are many ablation studies which allow readers to gauge the importance of different design decisions and components of the method presented in this paper.

**Weaknesses:**

- It would be interesting to see if the proposed method can be used train a single model jointly on all the modalities and benchmarks, instead of training one separate model for each. One of the limitations discussed by the authors is that the masking ratio can differ across modalities, which might pose a problem. However, building a general model that could simultaneously perform all the tasks could further demonstrate the generality of the proposed method.

**Questions:**

No questions.

**Limitations:**

To the best of my knowledge the authors have properly discussed the limitations of their work.

---

> ### Author Rebuttal · Authors · 2023-08-08
>
> Dear Reviewer b3vS,
>
> We sincerely thank you for your positive feedback and insightful comment to improve the manuscript. We address your comment below.
>
> ---
>
> **[Q1] Training a single model jointly on all the modalities and benchmarks to build a general model that could simultaneously perform all the tasks.** \
> **[A1]** Thank you for your insightful comment. We agree that it is a very interesting and important future direction for the modality-agnostic SSL research community to bind all modalities with a single model. Here, we believe MetaMAE can be quite a promising method to tackle this problem, e.g., by performing multiple modalities on a single model with domain-specific embedding modules as introduced in a recent work [1]. Furthermore, it is worth noting that MetaMAE outperforms other modality-agnostic SSL methods in paired multimodal datasets (see Table 1 in the attached pdf for MSCOCO results) where we believe this multimodal learning ability can help when unifying the modalities for SSL.
>
> [1] Zhang et al., Meta-Transformer: A Unified Framework for Multimodal Learning, Arxiv 2023

---

> > ### Comment · Reviewer_b3vS · 2023-08-16
> >
> > Thank you for your response. After reading the other reviews and the authors comments, I stick to my score and recommend acceptance.

---

> > > ### Author Response · Authors · 2023-08-16
> > > **Thank you for the response**
> > >
> > > Dear reviewer b3vS,
> > >
> > > Thank you for letting us know! We are delighted to hear that our rebuttal addressed your questions well.\
> > > Also, thank you for reading other reviewers' comments and giving a positive review of our paper.
> > >
> > > If you have any further questions or suggestions, please do not hesitate to let us know.
> > >
> > > Thank you very much,\
> > > Authors

---

### Official Review · Reviewer_2wfj · 2023-07-26

**Soundness:** 3 good
**Presentation:** 2 fair
**Contribution:** 3 good
**Rating:** 5
**Confidence:** 3

**Summary:**

This work interprets MAE as a meta-learning framework to formulate a modality-agnostic SSL framework. Specifically, they formulate the masked data and unmasked data (nearby tokens) as the support set and the query set, respectively. Moreover, the authors proposed to use gradient-based meta-learning on a decoder and task contrastive learning to optimize the encoder. Extensive experiments with ablation studies show the effectiveness of the proposed framework.

**Strengths:**

+ The idea of leveraging meta-learning and MAE to address modality-agnostic SSL is novel and interesting.
+ The design of using gradient-based meta-learning on a decoder and task contrastive learning seems to be effective.
+ The experiments on various modalities and the detailed user study are convincing.

**Weaknesses:**

- It is necessary to further clarify the relation between the proposed method (i.e., MAE+meta-learning) and the task (i.e., modality-agnostic SSL), that is the motivation for formulating MAE as a meta-learning paradigm. For example, what's the advantage of  the meta-learning paradigm for modality-agnostic SSL? In the paper, the authors only explain that their formulation can "improve the generalization", which is not specific.
- It would be better to discuss the potential advantages of the proposed method to solve multimodal SSL.

**Questions:**

Please address my concerns in the paper Weaknesses.

**Limitations:**

The authors have addressed the limitations and potential negative societal impact of their work.

---

> ### Author Rebuttal · Authors · 2023-08-08
>
> Dear Reviewer 2wfj,
>
> We sincerely thank you for your helpful feedback and insightful comments to improve the manuscript. We address your comments and questions below.
>
> ---
>
> **[Q1] Relationship between the proposed method (i.e., MAE+meta-learning) and modality-agnostic SSL.** \
> **[A1]** We would like to clarify that the essential challenge of modality-agnostic SSL is to develop a pretext task that requires limited inductive biases (i.e., modality-specific knowledge), due to the difference of each modality. To tackle this challenge, in this work, we introduce a novel combination of two modality-agnostic approaches, namely MAE and meta-learning.
>
> Specifically, by rethinking MAE’s task as a meta-learning framework, we can enhance MAE’s performance toward a modality-agnostic SSL framework. Remark that recent studies [1,2] have demonstrated the potential of MAE to evolve into a modality-agnostic SSL, indicating that enhancing MAE in a modality-agnostic manner is a promising direction for advancing modality-agnostic SSL. Furthermore, meta-learning, which is typically modality-agnostic, offers a promising direction for further improvement in a modality-agnostic manner.
>
> [1] Majmundar et al., MET: Masked Encoding for Tabular Data, NeurIPS 2022 Workshop TRL \
> [2] Xu et al., Masked Autoencoders that Listen, NeurIPS 2022
>
> ---
>
> **[Q2] Potential advantages of the proposed method to solve multimodal SSL.** \
> **[A2]** Thank you for the constructive suggestion. We indeed expect that our proposed MetaMAE can be further extended to handle multimodal SSL scenarios, e.g., by adapting recent works [1,2] on multimodal MAE. Table 1 in the attached pdf verifies the potential of MetaMAE toward multimodal as you suggested. We found that pretraining MetaMAE on MSCOCO (vision-language) dataset from the DABS benchmark outperforms other modality-agnostic SSL approaches for both the two vision-language downstream tasks (e.g., 57.6\% -> 69.7\% on VQA task). We thank the comment and will also discuss the potential of MetaMAE for multimodal datasets by including the result in the final manuscript.
>
> [1] Geng et al., Multimodal Masked Autoencoders Learn Transferable Representations, Arxiv 2022 \
> [2] Bachmann et al., MultiMAE: Multi-modal Multi-task Masked Autoencoders, ECCV 2022

---

### Official Review · Reviewer_GPRA · 2023-07-27

**Soundness:** 4 excellent
**Presentation:** 3 good
**Contribution:** 4 excellent
**Rating:** 8
**Confidence:** 4

**Summary:**

The paper titled "Modality-Agnostic Self-Supervised Learning by Meta-Learning" presents a novel framework called MetaMAE (Meta-learned Masked Auto-Encoder) for modality-agnostic self-supervised learning (SSL). The authors propose an innovative interpretation of Masked Auto-Encoder (MAE) as a meta-learning framework and enhance its generalization capabilities using advanced meta-learning techniques.

**Strengths:**

The paper includes extensive evaluations on multiple data modalities, demonstrating the effectiveness of MetaMAE. The results show significant improvements over previous modality-agnostic SSL methods in linear evaluation.

**Weaknesses:**

I do not see any critical weakness for this paper. Authors have done pretty good job working out the details and limitations.

**Questions:**

I guess it is more of a vague questions, but in your experience, what might help handling diverse and complex data with this approach? What is keeping MetaMAE to become the one approach to rule them all?

**Limitations:**

Authors have done a good job addressing the concerns and limitations on the paper.

---

> ### Author Rebuttal · Authors · 2023-08-08
>
> Dear Reviewer GPRA,
>
> We sincerely thank you for your positive feedback on our paper! We address your question below. If you have any other questions or suggestions, please let us know!
>
> Best, \
> Authors
>
> ---
>
> **[Q1] What might help in handling diverse and complex data with this approach? What is keeping MetaMAE to become the one approach to rule them all?** \
> **[A1]** We believe that our interpretation of MAE as a meta-learning framework will be helpful in learning more diverse and complex datasets. Since meta-learning is well-known for learning diverse and complex distributions of tasks to enhance the model’s generalization [1], we believe this interpretation would be helpful for learning complex data distributions (note that our key interpretation is to rethink data reconstruction as a task). Moreover, as recent studies highlight the potential of improving MAE for modality-agnostic SSL [2,3], we believe combining MAE with even more advanced meta-learning schemes (or developing a new meta-learning method) would be a promising and interesting future direction to explore.
>
> [1] Finn et al., Model-agnostic Meta-learning for Fast Adaptation of Deep Networks, ICML 2017 \
> [2] Majmundar et al., MET: Masked Encoding for Tabular Data, NeurIPS 2022 Workshop TRL \
> [3] Xu et al., Masked Autoencoders that Listen, NeurIPS 2022

---

> > ### Comment · Reviewer_GPRA · 2023-08-17
> > **Finalized comment from reviewer**
> >
> > Thank you for your response, I do not have any more concerns/questions. The score and acceptance stays the same. Best wishes!

---

> > > ### Author Response · Authors · 2023-08-17
> > > **Thank you for the response**
> > >
> > > Dear reviewer GPRA,
> > >
> > > Thank you for letting us know! We are happy to hear that our rebuttal addressed your questions well.\
> > > Also, thank you again for giving a positive review of our paper.
> > >
> > > If you have any further questions or suggestions, please do not hesitate to let us know.
> > >
> > > Thank you very much, \
> > > Authors

---

### Official Review · Reviewer_3Z8m · 2023-08-09

**Soundness:** 3 good
**Presentation:** 3 good
**Contribution:** 2 fair
**Rating:** 6
**Confidence:** 5

**Summary:**

This paper proposes MetaMAE, a novel framework to improve Masked Autoencoder (MAE) for modality-agnostic self-supervised learning. Unlike many SSL methods reliant on domain-specific inductive biases, MetaMAE enhances MAE with two meta-learning techniques: latent adaptation via gradient-based meta-learning to optimize amortized features and task contrastive learning to align amortized and optimized features. Conducting extensive experiments with modality-agnostic SSL benchmark DABS 1.0 and 2.0, MetaMAE consistently improves across diverse modalities like time series, tabular, images, and speech.

**Strengths:**

* **(S1)** This paper is well-motivated to enhance MAE with meta-learning strategies and design modality-agnostic pre-training methods. Extensive experiments on DABS and DABS 2.0 show the notable performance gains of the proposed MetaMAE, which highlights the potential of this research direction. Experimental details and the source code are provided for reproduction.

* **(S2)** The overall presentation is smooth and easy to follow. The background and related works of the manuscript are well-studied.

**Weaknesses:**

* **(W1)** The designed method is straightforward and simple, while the intuition or analysis of why meta-learning strategies improve MAE for modality-agnostic pre-training is not well studied. Since the vanilla masked predictive learning frameworks like MAE and Data2Vec [1] are already modality-agnostic, the author should provide some empirical analysis to support the statements in Sec. 3.2. Or it will decrease the novelty of MetaMAE.

* **(W2)** Some hyper-parameters need tuning for each modality, e.g., the decoder sizes and pre-training settings. As provided in the appendix, some sensitive hyper-parameters might have different behavior in different modalities.

* **(W3)** Although the authors study a wide arrange of related papers, some recently proposed general pre-training frameworks should be included. For example, Masked Auto-Encoder methods like Data2Vec variants [1, 2] (modality-agnostic) and SimMIM variants [3, 4] (architecture-agnostic); Multi-modality pre-training like BEiT variants [5]; Generative Pre-training like VQ-based variants [6, 7] (can be applied to various modalities with some modifications). It is better to discuss the similarity between MetaMAE and these methods.

* **(W4)** Typos should be double-checked before publishing, e.g., `pre-pre-training` in line 113.

#### Reference
[1] Alexei Baevski, et al. “data2vec: A General Framework for Self-supervised Learning in Speech, Vision and Language.” ICML, 2022.

[2] Alexei Baevski, et al. “Efficient Self-supervised Learning with Contextualized Target Representations for Vision, Speech and Language.” ICML, 2023.

[3] Chen Wei, et al. “Masked Feature Prediction for Self-Supervised Visual Pre-Training.” CVPR, 2022.

[4] Siyuan Li, et al. “Architecture-Agnostic Masked Image Modeling - From ViT back to CNN.” ICML, 2023.

[5] Wenhui Wang, et al. “Image as a Foreign Language: BEiT Pretraining for All Vision and Vision-Language Tasks.” arXiv, 2023.

[6] Patrick Esser, et al. “Taming Transformers for High-Resolution Image Synthesis.” CVPR, 2021.

[7] Tianhong Li, et al. “MAGE: MAsked Generative Encoder to Unify Representation Learning and Image Synthesis.” CVPR, 2023.

**Questions:**

Sorry for the late review! No more questions besides the concerns mentioned in Weaknesses.

**Limitations:**

Limitations have been discussed by the authors and I found no more limitations.

### Post-rebuttal
The authors' replies have tackled my concerns, thus, I raised the rating to 6. I hope the authors add more discussion with existing works and specify the limitations and future research directions for this topic.

---

> ### Author Rebuttal · Authors · 2023-08-10
>
> Dear Reviewer 3Z8m,
>
> We sincerely appreciate your efforts and insightful comments to improve the manuscript. We address your comment below.
>
> ---
>
> **[Q1] Why meta-learning improves MAE for modality-agnostic SSL?**\
> **[A1]** We remind that the masked modeling of MAE can be interpreted as an *amortization-based meta-learning* framework, where we improve the performance by incorporating advanced meta-learning schemes designed for enhancing amortization-based meta-learning. Typically, it is well-known that using i) gradient-based meta-learning [1] and 2) task contrastive learning [2] *on the amortized latent* enables the encoder to learn more complex and diverse task distribution. This is equivalent to learning complex and diverse data distribution from the MAE perspective, as individual data is interpreted as a meta-learning task (see Line 165). In this regard, we believe further using more advanced meta-learning for amortization-based meta-learning, e.g., MetaFun [3], will be an interesting and promising direction to explore.
>
> While we genuinely wished to provide supporting experimental results to validate these claims for addressing the reviewer's request, please understand that we only had **10 hours** (before the rebuttal deadline) to prepare our response (hence, implementing and running these experiments within the time limit was hard). Nonetheless, in the final manuscript, we will incorporate the results of using advanced meta-learning schemes, such as MetaFun, for MAE to further validate our claim.
>
> [1] Rusu et al., Meta-Learning with Latent Embedding Optimization, ICLR 2019\
> [2] Mathieu et al., On Contrastive Representations of Stochastic Processes, NeurIPS 2021\
> [3] Xu et al., MetaFun: Meta-Learning with Iterative Functional Updates, ICML 2020
>
> ---
>
> **[Q2] Some hyper-parameters differ across modalities.**\
> **[A2]** First, we would like to note that it is quite a common consensus that hyper-parameters differ across modalities, e.g., BERT (for text) uses a mask ratio of 0.15, and MAE (for image) uses 0.75. Nevertheless, we believe the performance of MetaMAE is robust to the hyper-parameter selection; we found even after sharing most hyper-parameters across *all modalities*, MetaMAE consistently outperforms previous state-of-the-art results across all modalities (see Table 3 in Appendix A). Furthermore, we run *additional* hyper-parameter ablation studies (see Table 2 in the attached pdf), which further support that the choice of pre-training hyper-parameters of MetaMAE is particularly robust at the intra-modality level. For example, for a Token modality, shared hyper-parameters $(\alpha, \lambda, \text{decoder depth}, P_\text{drop}) = (0.1, 0.01, 2, 0)$ can achieve 69.4\% on Genom and 61.5\% on Pfam, which are very close (or identical) to the reported performance of MetaMAE (69.4\%, and 62.3\%, respectively) in our original submission. These results could be useful observations in practice when pretraining a certain dataset of the same modality we experimented with.
>
> Finally, we also agree with the reviewer that developing a unified way of selecting hyper-parameters across modalities is an important and interesting direction. In this perspective, we believe MetaMAE can contribute as a strong baseline for future research in this direction, showing a robust performance regardless of the hyper-parameter selection (see Table 3 in the attached pdf); two major hyper-parameters ($\alpha, \lambda$) can be additionally shared across all modalities while it shows competitive performance with the MetaMAE in our original submission. We will include all results in the final manuscript.
>
> ---
>
> **[Q3] More discussion on the recent related works regarding general pre-training.**\
> **[A3]** Thank you for your suggestion. In the revised manuscript, we will include a comprehensive discussion of other pre-training methods that use masked modeling. Note that our key novelty of MetaMAE is to interpret the masked modeling (of MAE) as an amortization-based meta-learning framework, where such an interpretation is also feasible for other masked modeling schemes as well. While we have primarily chosen MAE due to its effectiveness across modalities and simplicity, we believe extending our interpretation for other masked modeling methods would be an interesting direction to explore by combining (or developing) advanced meta-learning frameworks.
>
> ---
>
> **[Q4] Typos?: “pre-pre-training”**\
> **[A4]** As for the word you mentioned, “pre-pre-training”, we note that this is not a typo but refers to a specific method proposed in [1]. Nevertheless, we will go through another proofreading to ensure that there are no further typos in the final draft.
>
> [1] Singh et al., The effectiveness of MAE pre-pretraining for billion-scale pretraining, Arxiv 2023

---

> > ### Comment · Reviewer_3Z8m · 2023-08-16
> >
> > Thanks for the detailed replies. I apologize again for my late review. Hopefully, the replies have addressed my concerns, and I raised the rating to 6. I hope the authors add more discussion with existing works and specify the limitations and future research directions for this topic.

---

> > > ### Author Response · Authors · 2023-08-16
> > > **Thank you for the response**
> > >
> > > Dear reviewer 3Z8m,
> > >
> > > We are very happy to hear that our response could help to address your concerns!
> > >
> > > In the revision, we will add more comprehensive discussions with existing works and also discuss the limitation and future directions as well.\
> > > Due to your valuable and constructive suggestions (based on your expertise), we do believe that our paper is much improved.
> > >
> > > If you have any further questions or concerns, please do not hesitate to let us know.
> > >
> > > Thank you very much,\
> > > Authors

---

### Author Rebuttal · Authors · 2023-08-10

Dear reviewers and AC,

We sincerely appreciate your valuable time and effort spent reviewing our manuscript.
As reviewers highlighted, we propose a novel (GRPA, 2wfj, Pe9M, 3Z8m) and interesting (2wfj, b3vS, 9KfC) approach with strong empirical results (ALL Reviewers) and extensive experiments (GPRA, 2wfj, b3vS, 9KfC, 8Z8m). Our paper is also well-structured (b3vS, 9KfC, Pe9M, 3Z8m).

We appreciate your constructive comments on our manuscript. In the attached pdf, we have run the following additional experiments to clarify the reviewers’ comments:

- Computational efficiency of pretraining MetaMAE (Figure 1; To answer the Q2 of Reviewer 9KfC)
- Large-scale multimodal experiments (Table 1; To answer the Q2 of Reviewer 2wfj, Q1 of Reviewer b3vS, and Q3 of Reviewer 9KfC)
- Shared hyperparameters across intra or all modalities (Table 2 and 3; To answer the Q1 of Reviewer Pe9M, and Q2 of Reviewer 3Z8m)
- More datasets on ablation studies (Table 4; To answer the Q3 of Reviewer Pe9M)

We strongly believe that MetaMAE can be a useful addition to the NeurIPS community, in particular, due to the enhanced manuscript by reviewers’ comments helping us better deliver the effectiveness of our method.

Thank you very much! \
Authors.

---

### Decision · Program_Chairs · 2023-09-21

**Decision:**

Accept (poster)

**Comment:**

This paper introduces a novel idea supported by strong empirical results. The reviewers appreciate the valuable contribution of this study.  In preparation for the camera-ready version, we kindly urge the authors to incorporate the feedback provided by the reviewers, which will undoubtedly further refine the quality and impact of the work.